# Analysis of Gene Regulatory Network and Transcription Factors in Different Tissues of the *Stropharia rugosoannulata* Fruiting Body

**DOI:** 10.3390/jof11020123

**Published:** 2025-02-07

**Authors:** Jia Lu, Jing Yan, Na Lu, Jiling Song, Jiayao Lin, Xiaohua Zhou, Xuebing Ying, Zhen Li, Zufa Zhou, Fangjie Yao

**Affiliations:** 1Hangzhou Academy of Agricultural Sciences, Hangzhou 310024, China; lujia920316@163.com (J.L.); yj2006@hz.cn (J.Y.); 13738068366@163.com (N.L.); songjiling860605@163.com (J.S.); ljymumu0215@126.com (J.L.); 2Engineering Research Center of Ministry of Education of China for Food and Medicine, Jilin Agricultural University, Changchun 130118, China; 3Tonglu Country Agricultural Technology Promotion Center, Hangzhou 311500, China; hztlzxh@163.com; 4Lin′an Agriculture and Forestry Technology Extension Center, Hangzhou 311302, China; 13588239832@163.com (X.Y.); lz-9701@163.com (Z.L.)

**Keywords:** *Stropharia rugosoannulata*, tissues, gene regulatory network, transcription factors, sections

## Abstract

*Stropharia rugosoannulata* is a mushroom that is rich in nutrients and has a pleasant flavor. Its cultivation area is expanding rapidly due to its simplicity and diversity. However, the developmental mechanism of the fruiting body, which constitutes the edible portion of *S. rugosoannulata*, remains to be elucidated. To address this knowledge gap, we conducted a comprehensive study. Our approach entailed the observation of sections through the fruiting body of *S. rugosoannulata* and the sequencing of the transcriptomes of various fruiting body tissues. The results demonstrated significant variations in the structure of the pileipellis, pileus, gill, veil, stipe, and trama of *S. rugosoannulata*. The predominant metabolic pathways included the amino acid metabolism of the pileus, sugar metabolism of the stipe, tryptophan metabolism, and wax production of the pileipellis, the DNA pathway of the gill, amino sugar metabolism of the veil, and the nitrogen metabolism of the trama. The promoter cis-element analysis revealed the roles of light response, methyl jasmonate, oxygen, and temperature on the differentiation of the veil, trama, and pileipellis, respectively. In summary, the present findings offer a molecular mechanism for the development of the fruiting body and provide directions for the enhancement of cultivation techniques of *S. rugosoannulata*.

## 1. Introduction

*Stropharia rugosoannulata*, also known as the wine cap Stropharia, is a member of the Strophariaceae family. *S. rugosoannulata* is distinguished by its pleasant flavor and high nutritional content, serving as a substantial source of protein, minerals, and biologically active compounds [1]. In light of these attributes, the Food and Agriculture Organization of the United Nations has identified it as a commendable food source for the development of nations [2]. The cultivation of *S. rugosoannulata* has emerged as a lucrative venture, attracting significant attention in China [3]. Since 2013, with the implementation of China’s precise poverty alleviation and rural revitalization strategies, the cultivation of *S. rugosoannulata* has become a priority project and has been vigorously promoted throughout the country. Statistical analysis reveals a substantial increase in the national production of *S. rugosoannulata*, with growth rates of 96.92%, 149.99%, and 40.67% from 2018 to 2020, respectively. Notably, in 2021, the production of fresh *S. rugosoannulata* mushrooms surpassed 410,500 tons, with large-scale cultivation becoming a reality in over a dozen provinces [4] *S. rugosoannulata* predominantly employs various agricultural and forestry byproducts, including straw, twigs, and mushroom residues, as a sustainable source of nutrients for mushroom cultivation, thereby contributing to the conservation of forest resources [5]. Furthermore, *S. rugosoannulata* demonstrates considerable promise in the domain of bioremediation. This is primarily due to its ability to degrade a wide range of environmental pollutants with diverse structures [6,7]. In addition, it has been observed to exhibit robust resistance to pathogens such as filamentous fungi and nematodes [8].

The process of fruiting body development is intricate, and its molecular mechanisms are not yet fully elucidated. Omics studies have facilitated the elucidation of the molecular mechanisms of *S. rugosoannulata*. Completing its genome has established a research foundation for the genetic dissection of significant traits and cultivar enhancement of *S. rugosoannulata*. The *S. rugosoannulata* genome is approximately 47.9–48.33 MB in size, with 20–21 scaffolds [2,5,9]. Transcriptomics can accelerate the study of the mechanisms of changes in the environment of *S. rugosoannulata*. For instance, the *S. rugosoannulata* transcriptome has been examined in response to low-temperature stress [10], high-temperature stress [11], Cd stress [12], humidity stress [13], and variations in nitrogen levels [14], among other mechanisms. The fruiting body, which is defined as an edible fungus, plays a critical role in the reproductive growth of fungi. Its development and maturation are orchestrated by a complex network of development-related genes and signaling pathways [15]. In a recent study, Wang C et al. utilized transcriptome analysis of tissues from different developmental periods of *S. rugosoannulata*, revealing that the development of the *S. rugosoannulata* fruiting body was mainly involved in biosynthesis, glucose metabolism, and amino acid metabolism [16]. In the analysis of the developmental transcriptomes across three distinct stages, Hao H et al. observed that the CAZyme gene exhibited its capacity to degrade the fruiting body not only during the mycelium stage but also potentially contributed to the formation of the cell wall during the fruiting body development phase. Carbonic anhydrase, which responds to changes in carbon dioxide concentration, in conjunction with its capacity to mediate the synthesis of cAMP and HSPs in response to changes in temperature, may also play a pivotal role in primordium formation and fruiting body development [10]. Li S et al. demonstrated through proteomic analyses that the protein expression profiles of the pileus and the stalk of the *S. rugosoannulata* fruiting body were distinct. The stalk exhibited a higher abundance of proteins implicated in biological processes associated with carbon metabolism, energy production, and stress response, while the pileus displayed a higher abundance of proteins involved in fatty acid synthesis and mRNA splicing [2].

Although prior studies have been conducted on the developmental process of the *S. rugosoannulata* fruiting body, the temporal and spatial specificity of gene expression during this process remains to be elucidated. This is primarily due to an inadequate number of samples from the various stages and tissues. To further enrich and strengthen the genomic resources of *S. rugosoannulata*, we employed high-throughput RNA-Seq to analyze the gene expression profiles of multiple tissues in the fruiting body of *S. rugosoannulata*, including the pileus, pileipellis, gill, veil, stipe, and trama tissues. The analysis of the gene regulatory networks of different tissue development and their constituent transcription factors has laid the foundation for elucidating the molecular mechanisms of edible mushroom fruiting body development. In summary, the in-depth transcriptome dataset developed in this study will serve as an important genomics resource for the discovery and characterization of genes that confer useful traits to *S. rugosoannulata*.

## 2. Materials and Methods

### 2.1. Sectioning and Observation of S. rugosoannulata Fruiting Body

The *S. rugosoannulata* strain S18 was deposited in the Hangzhou Academy of Agricultural Sciences. To obtain samples of different tissues of *S. rugosoannulata* fruiting bodies, cultivation experiments were carried out at the Institute of Agricultural Science of Hangzhou City. Samples of fruiting bodies were collected during the production of mushrooms. The entire fruiting body was preserved in FAA fixative (Servicebio, Wuhan, China) at ambient temperature for 24 h. Following fixation, the test material underwent a dehydration process in a dehydrator (DIAPATH, Milan, Italy) employing a sequential gradient of alcohol (75% alcohol for 4 h, 85% alcohol for 2 h, 90% alcohol for 2 h, 95% alcohol for 1 h, anhydrous ethanol I for 1 h, and anhydrous ethanol I for 1 h). Subsequently, the samples were dried in the dehydrator. The samples were then subjected to a dehydration process involving an initial 1-h treatment with anhydrous ethanol I, followed by 30 min of anhydrous ethanol II. This was followed by a 5–10 min soak in alcohol benzene and subsequent washes with xylene I and xylene II for an additional 5–10 min. The samples were then heated to 65 °C for 1 h to melt paraffin I. The samples were then heated to 65 °C for 1 h to melt paraffin II (1 h at 65 °C), followed by paraffin III (1 h at 65 °C). After the embedding, sectioning, and dewaxing of the tissue, it was stained with Toluidine Blue Stain (Servicebio, Wuhan, China) for 2 min. Subsequently, the tissue underwent a series of processing steps, including washing, baking, and transparentization with xylene for five minutes. Neutral gum sealing was then performed, and the tissue sections were scanned using a digital scanner (3DHISTECH, Budapest, Hungary) to capture the synthetic images. The resulting images were then examined using CaseViewer 2.4 (3DHISTECH, Budapest, Hungary) software, and the images were subsequently interposed.

### 2.2. RNA Extraction and High-Throughput Sequencing

Samples of the fruiting bodies were collected during the process of mushrooming and subsequently divided into six different tissues, including the pileipellis (the outer layer of the umbrella-like portion of the upper part of the fruiting body), pileus (the umbrella-like portion of the upper part of the fruiting body with the pileus skin removed), gill (ridges on the inner side of the pileus), stipe (the supporting portion of the pileus of the fruiting body), trama (the loose portion of the inner part of the stipe), veil (the lamellar membrane-like structure connecting the pileus and the stipe), pith (the inner fluffy part of the stalk), and mycelium (the lamina-like structure connecting the pileus to the stalk) (Figure 1). Total RNA was extracted from each sample using the RNA Prep Pure Plant Kit (Tiangen, Beijing, China). The RNA concentration and purity were measured using a NanoDrop 2000 (Thermo Fisher Scientific, Wilmington, DE, USA). The RNA integrity was then assessed using the RNA Nano 6000 assay kit on an Agilent Bioanalyzer 2100 system (Agilent Technologies, Santa Clara, CA, USA). Subsequently, sequenced cDNA libraries were generated using the Hieff NGS Ultima Dual-mode mRNA Library Prep Kit for Illumina (Yeasen Biotechnology (Shanghai) Co., Ltd., Shanghai, China). The libraries were then subjected to sequencing on the Illumina NovaSeq platform(Illumina, Inc., San Diego, CA, USA), per the manufacturer’s instructions, to yield 150 base pairs of double-terminal sequences.

### 2.3. RNA-Seq Readings Are Mapped to the Reference Genome and the Genes Are Re-Annotated

A total of 18 samples from diverse tissues were subjected to RNA-seq analysis, and the resulting reads were processed using an in-house Perl script (Beijing Biomarker Technologies Co., Ltd., Beijing, China). This process entailed the removal of sequences containing junctions, sequences containing poly-N, and low-quality sequences. Subsequently, the Clean Reads were aligned to the reference genome, *S. rugosoannulata* reference genome QGU27 (GenBank Registry Number JANCIK000000000.1), using Hisat2 [17]. Subsequently, the aligned reads were assembled using StringTie 2.2.2 [18], which reconstructed the transcriptome for further analysis. The spliced transcripts were then compared with the original genome annotation information to find the original unannotated transcribed regions. This process was undertaken to augment and refine the existing genome annotation information. Finally, the spliced transcripts were compared with the Non-Redundant Protein Sequence Database (NR) database [19], the euKaryotic Orthologous Groups (KOG) database [20], the Kyoto Encyclopedia of Genes and Genomes (KEGG) database [21] and the Gene Ontology (GO) database [22] to obtain the annotation information of new genes. The predicted genes were then used for transcription factor prediction using iTAK 18.12 software [23].

### 2.4. Principal Component Analysis (Pca), Hierarchical Clustering, and Sample Distance

The StringTie maximum flow model was used to evaluate gene expression, and FPKM (Fragments Per Kilobase of transcript per Million fragments mapped) was used for standardization. The FPKM density distribution and FPKM distribution box plot between samples were further plotted to analyze the distribution characteristics and dispersion degree of gene expression in different tissues. Spearman’s Correlation Coefficient was employed as an assessment index of biological replicate correlation [24] and Principal Component Analysis (PCA) was executed using the R scatterplot3d package to globally compare the relative correlations of samples from different tissues. Three-label plots of PC1, PC2, and PC3 were generated using the R ggplot2 package.

### 2.5. Gene Co-Expression Analysis and WGCNA Network Construction

To identify genes demonstrating analogous expression patterns in disparate tissues, a weighted gene co-expression network analysis (WGCNA) [25] was conducted, employing the entire set of duplicated expressed genes. The identification of closely linked groups of genes with similar expression patterns was facilitated through the implementation of the gene module concept. The soft threshold capacity was determined by the pickSoftThreshold function in the package, and the “signed hybrid” network model was used. Hierarchical clustering was then performed based on the topological overlap matrix, and the generated dendrograms were cut using the dynamic tree cut program. The final network was constructed using the blockwiseModules function with parameters minModuleSize = 30, MaxModuleSize = 5000, and mergeCutHeight = 0.2. The determination of specific gene modules with a higher correlation between different correlations of tissue expression was made based on significantly higher Pearson’s correlation coefficient values (r > 0.8, *p*-value < 0.001).

### 2.6. Biological Processes and KEGG Analysis of Genes

The GO and KEGG pathways were enriched using the R clusterProfiler package [26]. All expressed genes were utilized as background genes. Subsequent filtration of enriched GO terms was conducted according to the “medium similarity” parameter, employing REViGO analysis (http://revigo.irb.hr, accessed on 29 July 2024).

### 2.7. Analysis of Tissue-Specific Expressed Genes and Transcription Factor Regulation

The calculation of the TAU (tissue-specific gene expression) index was performed on the tissue-specific expressed gene module of the WGCNA results to determine the different tissue-specific expressed genes [27]. The R pheatmap 1.0.12 software package was then utilized to cluster all specific genes, followed by heat mapping and Z-score conversion, before the heat mapping process. The prediction of hypothetical targets regulated by different transcription factors was conducted using GENIE3 [28] with TFs designated as query regulators, employing the following parameters: tree.method = “RF”, ntrees = 1000, threshold = 0.003. The obtained targets were documented.

### 2.8. Analysis of Cis-Acting Elements of Tissue-Specific Transcription Factor Promoter

A tissue-specific expression analysis of the transcription factor promoter cis-acting element was conducted using PlantCARE (http://bioinformatics.psb.ugent.be/webtools/plantcare/html/, accessed on 10 August 2024). Subsequently, the analysis was represented visually through the utilization of TBtools.

## 3. Results

### 3.1. Observation on Fruiting Body Tissue of S. rugosoannulata

A thorough investigation was conducted on paraffin sections of *S. rugosoannulata* fruiting bodies, unveiling the existence of six distinct tissues: pileus, pileipellis, gill, veil, stipe, and trama. This observation was corroborated by the results of the sections, in which the cell walls were thicker in the stipe tissue and the gill tissue. The pileus and pileipellis tissues were interspersed with densely arranged cells characterized by thick cell walls, and the cells from the pileus to the veil tissues were tightly to loosely packed (Figure 2a–c). The pileipellis cells manifest transversely oriented arrangements, accompanied by gap structures exhibiting variability in size (Figure 2d). The pileus tissue manifests as irregularly kinked and dense (Figure 2e). The outermost layer of the gill is characterized by stromatocytes separating gills, and basidiospores are ejected from basidia between stromatocytes. The inner part of the gill is marked by regular longitudinal arrangements (Figure 2f); the veil tissue is irregularly kinked and loosely arranged (Figure 2g); and the stipe tissue is similar to but loosely arranged vascular cells with small intermediate gap structures (Figure 2h). The stipe tissue is composed of longitudinally arranged vascular cells with small interstitial spaces, and the trama cells are similar to the stipe cells but loosely arranged and with less thickening of the cell walls (Figure 2h,i).

### 3.2. Genome Re-Annotation of S. rugosoannulata

To investigate the key metabolic pathways in different fruiting body tissues of *S. rugosoannulata* S18, high-throughput RNA sequencing of the pileus, pileipellis, gill, veil, stipe, and trama (Figure 1) was conducted, leading to a total of 18 RNA-seq libraries, three from each tissue. The analysis yielded a total of 414.19 pair-end reads and 123.78 gigabytes of clean data from these tissues. Genome and annotation files are critical for functional gene studies. A comprehensive investigation was conducted to ascertain published genome data for *S. rugosoannulata*, culminating in the identification of three genome data sets (Appendix A). The genome of QGU27 was selected as the reference genome due to its recent publication and the availability of assembly and annotation information. The BUSCO score for this genome was 98.7%, indicating a state of high completeness [9]. To obtain higher-quality annotation information for the *S. rugosoannulata* genome, the original annotation information was corrected using the 18 RNA-seq libraries from this study. Consequently, a novel database (v.2.0) comprising 17,938 predicted protein-coding genes was formulated, underpinned by the RNA sequence mapping gene model (Figure 3 and Appendix A). A total of 2212 genes were incorporated into version 2.0, representing an augmentation of 2212 genes compared with version 1.0 (Figure 3). The genes that were newly incorporated were designated with the prefix “newgene_xx”. The fundamental statistical characteristics of versions 1.0 and 2.0 are delineated in Table 1.

To enhance comprehension of the functional annotations present within the v.2.0 database, a comprehensive search was conducted, encompassing all protein sequences and integrating various databases, including GO, KEGG, KOG, iTAK, and others. Among the 17,938 transcripts examined, 8369 transcripts (46.65%) were categorized into specific GO terms, which surpasses the 6779 transcripts categorized into specific GO terms in v.1.0. Additionally, 6315 transcripts (35.2%) were categorized into specific KEGG pathways, which exceeds the 4063 transcripts categorized into specific KEGG pathways in v.1.0; 4675 transcripts (26.06%) were categorized into specific KOG systems, which was superior to the 4231 categorized transcripts in v.1.0 (Table 1 and Appendix A). Furthermore, the identification and categorization of transcription factors/regulators and protein kinases was conducted using the iTAK software, resulting in the identification of 397 transcription factors/regulators and 164 protein kinases in the v.2.0 annotation (Figure 3 and Appendix A).

### 3.3. Analysis of Gene Expression in Different Developmental Tissues of S. rugosoannulata

To make the gene expression levels of different genes and samples comparable, we employed FPKM to normalize the expression amount. As demonstrated in the box line plot (Figure 4a), the distribution of gene expression was essentially equivalent among the samples, except for gill tissues, which exhibited a marked increase compared to the other tissues. The density distribution plot of FPKM (Figure 4b) revealed that the density distribution curves of the 18 sequenced samples exhibited greater consistency, yet the gene expression density of gill tissue was considerably higher than that of the other tissues. To achieve a comprehensive understanding of the differences between the developmental tissues of *S. rugosoannulata*, an analysis was conducted in which expressed genes with FPKM > 0.01 were screened. The analysis yielded a total of 10,559 expressed genes, accounting for 67.3% of the total genes. The results of the principal component analysis and hierarchical clustering demonstrated a clear clustering of duplicate samples representing the same tissues (Figure 4c,d). Subsequent analysis of the correlation between all the samples revealed the capacity to categorize them into three groups, with gill tissues comprising the first group; a second group consisting of stipe and trama tissues; and three groups of pileipellis, veil, and pileus tissues (Figure 4d). The 100 most highly expressed genes were mapped (Figure 4e), and the results demonstrated that the expression values of the majority of genes ranged from 0.08 to 29,700, and these genes exhibited distinct tissue-specific expression patterns. For instance, 20 genes exhibited elevated relative expression levels in the pileipellis, 10 genes demonstrated high relative expression levels in the gill, 26 genes displayed high relative expression levels in the pileus, 10 genes showed high relative expression levels in the stipe, 13 genes manifested high relative expression levels in the trama, and 21 genes revealed high relative expression levels in the veil (Figure 4e). This finding collectively suggests that there exists spatial diversity in gene expression within the fruiting body of *S. rugosoannulata*.

### 3.4. Gene Co-Expression Network of S. rugosoannulata

To ascertain the mechanism of differentiation in different tissues of *S. rugosoannulata*, WGCNA was utilized to analyze the 10,559 expressed genes in the RNA-seq dataset. This analysis yielded clusters of genes that differed in 16 modules, whose expression covered six different tissues (Figure 5a). Of particular interest are the six modular genes that exhibited specificity for different tissues in the development of the *S. rugosoannulata* fruiting body. Specifically, MEblue, MEbrown, MEgreen, MEgrey60, MEturquoise, and MEyellow exhibited predominant expression in the pileipellis, trama, stipe, pileus, gill, and veil tissues, respectively (Figure 5b). Pearson correlation coefficient values (r) and ME values were utilized to ascertain the relationship of modules to specific tissues. The findings revealed a strong correlation between MEgrey60 and pileus tissue (r = 0.97, *p* = 0.002); MEblue demonstrated a significant correlation with pileipellis tissue (r = 1, *p* = 2 × 10^−5^); MEyellow exhibited a significant correlation with veil tissue (r = 1, *p* = 3 × 10^−5^); MEturquoise exhibited a significant correlation with gill tissue (r = 1, *p* = 1 × 10^−5^); and MEbrown revealed a significant correlation with trama tissue (r = 0.89, *p* = 0.02). A statistically significant correlation was identified between MEgreen and stipe tissue and gill tissue (r = 0.98, *p* = 6 × 10^−4^) (Figure 5c). Furthermore, the absence of tissue-specific expression patterns in the remaining modules indicates the potential complexity of the regulatory mechanisms governing these gene clusters. Consequently, our co-expression analysis identified distinct tissue-specific genes and gene clusters exhibiting analogous expression patterns, thereby establishing a framework for investigating fundamental biological processes in *S. rugosoannulata*.

### 3.5. Biological Processes and KEGG Analysis of Genes in Different Tissues of S. rugosoannulata

A comprehensive investigation was conducted to elucidate the potential biological processes associated with diverse *S. rugosoannulata* tissues. To this end, a multifaceted analytical approach was employed, encompassing GO enrichment analysis and KEGG metabolic pathway analysis of tissue-specific expressed gene modules. The analysis of tissue-specific genes revealed that different gene modules exhibited similarities in the enrichment process, including metabolic processes, membrane composition, and catalytic function (Appendix A). To further investigate the specific biological processes occurring in various tissues, deinterlacing operations were performed on biological processes that were found to be significantly enriched in different gene modules. The results of this analysis revealed the presence of 13, 6, 6, 6, 52, 12, and 16 specific pathways in the pileus, pileipellis, veil, gill, trama, and stipe, respectively (Figure 6a and Appendix A). REViGO analysis revealed that the primary biological processes encompassed by pileus tissue included metabolic processes, positive regulation of GTPase activity, carboxylic acid biosynthetic processes, and purine-containing compound metabolic processes, among others. In contrast, the predominant biological processes observed in pileipellis tissue included sulfur amino acid metabolic processes and responses to oxidative stress. Furthermore, the predominant biological processes observed in veil tissue included regulation of transcription by RNA polymerase I, endocytosis, homologous recombination, and L-methionine biosynthetic processes. Notably, gill tissues exhibited the regulation of DNA replication as their predominant biological process, along with additional processes such as translation, chromosome segregation, maintenance of location, and protein-DNA complex assembly. In the case of trama tissue, the predominant biological processes include the G2/M transition of the mitotic cell cycle, intracellular monoatomic cation homeostasis, and positive regulation of phosphorylation. Conversely, the predominant biological processes observed in stipe tissue encompass protein transport, the NAD metabolic process, positive regulation of response to stimulus, and the glucosamine-containing compound metabolic process (Figure 6b). We proceeded to identify tissue-specific pathways by conducting a KEGG analysis, which revealed that tryptophan metabolism, cutin, suberin, and wax biosynthesis are tissue-specific pathways in pileipellis; amino sugar and nucleotide sugar metabolism are specific pathways in veil tissues; and ribosome, DNA replication, proteasome, homologous recombination, nucleotide excision repair, and mismatch repair are tissue-specific pathways in gill tissue. Glycerophospholipid metabolism and starch and sucrose metabolism are specific modules of stipe organization; nitrogen metabolism and indole diterpene alkaloid biosynthesis are specific modules in trama organization; phenylalanine, tyrosine, and tryptophan biosynthesis are specific modules in pileus (Figure 6c). The utilization of GO and KEGG terms serves to provide further support for the functional annotation of clusters of genes expressing key biological processes and metabolic pathways in different tissues of the *S. rugosoannulata* fruiting body.

### 3.6. Tissue-Specific Expression of Transcription Factors in S. rugosoannulata

The identification of dominant genes in different tissues can provide resources to determine the function of these genes in their respective differentiation and crosstalk between tissues. Subsequent calculations of the tissue-specific genes analyzed by WGCNA using the TAU index revealed that gill, pileipellis, veil, trama, stipe, and pileus tissues exhibited predominant expression (TAU > 0.85) of 1415, 486, 320, 272, 173, and 54 genes, respectively (Figure 7a and Appendix A). Transcription factors are critical molecules that regulate gene expression, determining the spatio-temporal expression of genes. A total of nine distinct transcription factors were identified as being specifically expressed in the gill tissue. Of these, five genes belonged to the C2H2 family of transcription factors, two genes belonged to the HB-other family, and the remaining two genes belonged to the C2C2-YABBY and ZN-CULS families, respectively. Pileipellis and pileus specifically expressed one transcription factor each, both belonging to the C2H2 family; one specifically expressed transcription factor each in stipe and trama, both belonging to the Zn_clus gene family; and two specifically expressed transcription factors in veil belonging to the C2H2 and C2C2-GATA families, respectively (Figure 7b). The dynamic expression patterns of genes are indicative of their roles in different developmental tissues. To identify such expression changes, we performed transcription factor co-expression analyses of their corresponding targets using an integrated approach based on the GENIE3 tool. The analysis revealed that each tissue-specific expression TF predicted between 36 and 821 potential target genes (Appendix A). The 10 genes with the highest correlation for each transcription factor were mapped, and it was found that the targets predicted by *NewGene_2207*, all of which are specifically expressed in pileus tissues, and the targets predicted by *SrGene12752*, all of which are specifically expressed in stipe. *SrGene00005* predicted targets were found to be specifically expressed in trama. *SrGene02956* and *SrGene11048* predicted targets were found to be specifically expressed in veil tissues, and *SrGene02956* regulates *SrGene11048*. *SrGene12185* predicted genes were found to be specifically expressed in pileipellis. Nine distinct transcription factors in gill form four networks, of which *SrGene14904*, *SrGene12874*, *SrGene2123*, *SrGene03412*, and *SrGene06553* form one network, *SrGene11430* and *SrGene01859* form one network, and *SrGene11590* and *SrGene03418* form one network each (Figure 7c). In summary, we conclude that tissue-specific expression of transcription factors is a key factor in the differentiation of different tissues.

### 3.7. Analysis of Cis-Acting Elements of Tissue-Specific Transcription Factors in S. rugosannulata

To investigate the potential biological functions of distinct tissue-specific transcription factors, cis-acting elements in the 2000 base pairs (bp) region upstream of the gene promoter were identified using the PlantCARE tool (Figure 8). The promoter region of *NewGene_2207* contains cis-elements associated with light responsiveness, methyl jasmonate (MeJA) responsiveness, abscisic acid responsiveness, circadian control, and low-temperature responsiveness. In contrast, *SrGene00005* manifests MeJA-responsiveness, light responsiveness, abscisic acid responsiveness, and auxin responsiveness within its promoter region. Furthermore, the *SrGene12752* promoter region contains cis-elements for light responsiveness, abscisic acid responsiveness, MeJA-responsiveness, anaerobic induction, low-temperature responsiveness, gibberellin responsiveness, and salicylic acid responsiveness. The promoter regions of both *SrGene02956* and *SrGene11048* contain cis-elements of light responsiveness, MeJA-responsiveness, abscisic acid responsiveness, low-temperature responsiveness, and anaerobic induction. Conversely, *SrGene012185* exhibits a distinct profile, containing light responsiveness, MeJA-responsiveness, abscisic acid responsiveness, anaerobic induction, and low-temperature responsiveness. The promoters of genes specifically expressed in gill predominantly contain cis-elements associated with gibberellin responsiveness, light responsiveness, MeJA-responsiveness, abscisic acid responsiveness, anaerobic induction, auxin responsiveness, low-temperature responsiveness, gibberellin responsiveness, salicylic acid responsiveness. The collective analysis of these results indicates that transcription factors originating from disparate tissues assume pivotal functions in the growth and development of *S. rugosoannulata*, as well as in its response to adverse environmental conditions.

## 4. Discussion

### 4.1. S. rugosoannulata Fruiting Body Has a Different Cellular Tissue

To achieve a more precise distinction between the various tissues of *S. rugosoannulata*, we performed paraffin sections of the fruiting body of *S. rugosoannulata* and stained them with toluidine blue. Our analysis revealed the presence of two distinctly different types of tissues throughout the stipe: the outer stipe tissues exhibited a tightly arranged longitudinal configuration as vascular bundles, while the inner trama cells demonstrated a more loosely arranged arrangement. The cell wall thickness of the stipe tissue was higher than those of the trama tissue, which might be the reason for the occurrence of stalk dehiscence in *S. rugosoannulata* under different humidity conditions [13]. During the development of *Agaricus bisporus*, the cells of the stipe adopt a long, columnar shape, characterized by a more compact arrangement of cells. The synthesis and arrangement of cellulose, chitin, and other components of the cell wall contribute to the cell wall’s optimal strength and flexibility [29]. In the present study, we observed the presence of densely arranged cells with thick cell walls in the cap of the *S. rugosoannulata* tissues, specifically the pileus and pileipellis. This finding contributes to our understanding of the family’s capacity to differentiate between the characteristics of the pileus and pileipellis [30]. In the present study, it was ascertained that the trama and veil tissues exhibited substantial voids within their reticulation structures. These findings suggest that autolysis occurs in these tissues, a hypothesis that is also validated by the hollowing phenomenon found in the middle of the stipe during cultivation [31].

### 4.2. An Accurate Genetic Model Will Contribute to the Functional Genomics Research of S. rugosoannulata

The accuracy of gene model annotation is instrumental in the identification of genes, the construction of gene expression profiles, and the execution of molecular biology experiments [32]. In previous studies, gene models were primarily annotated using de novo prediction, homology-based prediction, and transcriptome-based prediction using computer software. However, these methods have been shown to contain numerous prediction errors in the annotation files. To address these limitations, researchers have employed the re-annotation of more accurate genomic and transcriptomic data in several studies, including those on *Hericium erinaceus* [33], *Ganoderma lucidum* [34], and *Sanghuangporus sanghuang* [35]. In this study, adjusted gene models based on 18 RNA-Seq datasets (including 6 tissues) were published to map RNA-Seq data to the genome from de novo annotation to 2212 genes, with a total of 17,938 individual genes annotated, of which 10,559 (58.9%) transcripts were expressed at levels higher than 0.01 FPKM and 7379 (42.1%) non-expressing transcripts. It is plausible that the expression of certain genes may be observed in tissues or conditions that have not yet been examined; consequently, additional RNA-Seq data from diverse tissues is required to enhance the precision of these gene models. For instance, the *Phellinus gilvus* study facilitated the identification of additional genes with medicinal functions in the dikaryon mycelium transcriptome data [36].

### 4.3. Threshold Determination of Gene Expression in Different Tissues of S. rugosoannulata

FPKM was utilized to normalize the expression of diverse genes, revealing that the expression density of genes in gill tissues was notably higher than in other tissues. This finding was further substantiated by the box-and-line plot density distribution of gene expression in different tissues, indicating that gill tissues express genes at potentially higher levels than other tissues. Furthermore, the analysis of differentially expressed genes revealed their capacity to modify the distribution of the overall expression level of all genes, a phenomenon that is consistent with the observation that gill-specific expression of genes is significantly higher than in other tissues, as previously verified. Gene expression FPKM < 1 is generally designated as non-expressed genes. As demonstrated in Figure 4a,b, a considerable proportion of genes exhibit FPKM < 1 across diverse tissular contexts. Consequently, the filtration of these genes might lead to an inaccurate analysis. Therefore, the selection of genes with FPKM > 0.01 as expressed genes was deemed appropriate. The subsequent principal component analysis and hierarchical clustering of these expressed genes yielded a clear clustering of tissue gene expression, which was divided into three groups: the gill group; the stipe and trama group; and the pileipellis, veil, and pileus group. This division corresponded to the apparent morphology and sectioned tissues.

### 4.4. Different Tissue-Specific Expression Genes and Main Functions of S. rugosoannulata

WGCNA can be used to screen genes that are highly correlated with target traits, obtain a series of biologically significant co-expression modules, and identify key genes behind these traits [37]. WGCNA has been employed in diverse areas of biology due to its precision and effectiveness as a bioinformatics and biological data mining instrument [37]. This includes the identification of tumor tissue markers [38] and the study of plant and fungus-specific tissue-expressed genes [39,40]. A thorough investigation of the genes expressed in various tissues was conducted, employing WGCNA, a process that culminated in the identification of a total of 16 modular genes. Of these 16 modular genes, 6 modules were found to be highly expressed in different tissues and demonstrated a high degree of correlation. Our enrichment analysis of the modules expressed in different tissues revealed that the pileus-specific biological processes and metabolic pathways included the carboxylic acid biosynthetic process and the purine-containing compound metabolic process. Stipe-specific biological processes and metabolic pathways included glucosamine-containing compound metabolic process, glycerophospholipid metabolism, and starch and sucrose metabolism. These findings were consistent with the previously reported higher content of crude protein, total amino acid, crude fat, organic acid, and nucleotide in the fruiting body of *S. rugosoannulata*, as compared to the stipe [41]. The primary biological processes that are distinct to pileipellis include sulfur amino acid metabolic processes, tryptophan metabolism, cutin, suberin, and wax biosynthesis, among others. Studies have revealed that the tryptophan metabolite 3-hydroxykynurenine undergoes rapid oxidation by molecular oxygen at neutral pH, resulting in the formation of multiple colored products [42]. This observation may be pertinent to the variation in the color of the caps of *S. rugosoannulata*. The metabolism of tryptophan, cutin, suberine, and wax biosynthesis is reflected in *S. rugosoannulata* pileipellis, which is characterized by distinct scales on the upper part. The biological and metabolic processes of gill are associated with DNA biosynthesis, including homologous recombination, nucleotide excision repair, and mismatch repair, among others. A study of *G. lucidum* spore formation revealed the presence of the same mismatch repair pathway [43]. Our research has revealed that nitrogen metabolism constitutes a distinct expression pathway in *S. rugosoannulata*. Prior studies have demonstrated that an augmented supply of nitrogen can distinctly enhance the rate of high-quality mushrooms in *S. rugosoannulata*. The presence or absence of pith has been identified as a determining factor in the rate of high-quality mushrooms in *S. rugosoannulata* [44].

### 4.5. Transcription Factors Regulate the Development of Different Tissues of the S. rugosoannulata Fruiting Body

Transcription factors play an integral role in signaling and promoting growth, development, and metabolism in macrofungi. In recent years, an increasing number of research results have been employed to explore the relationship between development and transcription factors in macrofungi. In this study, a combination of WGCNA and TAU indices was utilized to identify the transcription factors that were predominantly expressed in various tissues. This approach ultimately led to the identification of 15 genes that were predominantly expressed in different tissues and were found to belong to five distinct gene families: C2H2, C2C2-YABBY, C2C2-GATA, HB-other, and Zn-clus. The C2H2 family of genes has been observed to be expressed in the pileus, gill, and veil, as well as the pileipellis. Genes belonging to the C2H2 family have been found to regulate accelerated fruiting body formation, mediate spore formation, and perform related functions [45]. Ohm, A. et al. found that the C2H2 gene was significantly up-regulated during fruiting body formation. After the knockout of the gene, the development of the dikaryotic strain stagnated at the aggregate formation stage [46]. The C2C2-GATA family, SrGene02956, is involved in nitrogen control, iron carrier biosynthesis, circadian regulation, and mating type switching [47]. Studies have found that C2C2-GATA is up-regulated during primordium development. Knockout of this gene causes morphological changes in the fruiting bodies of dikaryotic strains to form more but smaller mushrooms [46]. The C2C2-YABBY family member *SrGene06553* is expressed exclusively in the gill, and studies have demonstrated that YABBY transcription factors have been identified as playing a pivotal role in the formation and development of plant reproductive organs [48]. HB-other family members *SrGene03418* and *SrGene11590* are expressed at high levels in gill tissue. Prior research has elucidated that the Homeobox transcription factor exerts a regulatory influence on hyphal production [49]. The zinc-cluster family members *SrGene00005* and *SrGene12752* are expressed in the pith and stipe, respectively. Ohm, A. et al. found that zinc-cluster could inhibit the formation of primordium, and knockout of this gene could not form primordium and mushroom [46]. Previous studies have demonstrated the critical function of zinc cluster proteins in stipe elongation [50].

### 4.6. Tissue-Specific Expression of Transcription Factors in S. rugosoannulata Plays an Important Role in Environmental Response

Cis-acting elements are defined as DNA sequences that possess regulatory activity, thereby controlling gene expression and playing a crucial role in development and physiology [51]. These elements ensure the appropriate spatiotemporal pattern of gene expression, which is imperative for normal development and environmental responses [52]. In this study, we sought to predict cis-acting elements in the promoter regions of five transcription factor families, encompassing a total of 15 genes, which are expressed dominantly in six distinct tissue species. Our analysis identified elements associated with abiotic stresses and phytohormones. The results indicated that light responsiveness response elements were the most prevalent, followed by the MeJA element, and hormone-responsive progenitors such as growth hormone, salicylic acid, abscisic acid, and gibberellin. The physiological and environmental cis-acting elements were also found in the promoters of the fungal immunomodulatory protein in *Flammulina velutipes* and the *hat* gene in *Cordyceps militaris* [53,54]. Light has been demonstrated to play a pivotal role in the growth and development of edible mushrooms. Im et al. observed that light color significantly impacts the coloration of *F. velutipes* [55]. In recent years, it has been found that phytohormones also have significant physiological effects on edible mushrooms. MeJA, a multifaceted molecule, has been shown to respond to various environmental stresses, including salt stress, drought, and low temperature. This response has been found to enhance plant resilience and survival through various mechanisms. In a related study, Dai et al. found that supplementation with MeJA induced a significant induction of ergosterol production in *H. erinaceus* [56]. Furthermore, they observed that MeJA supplementation triggered a defense reaction in the mycelium of the monkey head mushroom [56]. Ramachella et al. found that growth hormones could promote the growth of the cap of flat mushrooms [57]. Ji et al. found that low concentrations of salicylic acid promoted the growth of flat mushrooms, while high concentrations of salicylic acid inhibited growth [58]. Cui et al. discovered that abscisic acid could effectively promote the accumulation of triterpenes in G. lucidum [59]. Li et al. reported that the dynamics of gibberellin content were closely correlated with the phenomenon of stipe elongation in *F. filiformis* [60]. It is noteworthy that the transcription factor *NewGene_2207*, which is expressed in pileus tissues, contains circadian control response elements, suggesting sensitivity to circadian changes and a potential role in regulating circadian adaptive mechanisms. *SrGene02956* contains 17 light response elements, suggesting high sensitivity to light response and a potential role in veil differentiation. *SrGene00005* contains 16 MeJA elements, indicating a high sensitivity to methyl jasmonate changes and a potential role in trama differentiation. *SrGene12185* contains four anaerobic induction and three low-temperature responsiveness elements (Appendix A), suggesting sensitivity to oxygen and temperature changes and a potential role in the differentiation of pileipellis. In summary, we posit that dominantly expressed transcription factors in different tissues of *S. rugosoannulata* may play an important role in environmental response. By modulating stress physiology and hormonal signaling, these genes promote the differentiation of diverse tissues within *S. rugosoannulata*.

## 5. Conclusions

Amino acid metabolism, sugar metabolism, tryptophan metabolism and wax production, DNA pathway, amino sugar metabolism, and nitrogen metabolism affect the differentiation of pileus, stipe, pileipellis, gill, veil, and trama, respectively. Light reaction, methyl jasmonate, oxygen, and temperature affect the differentiation of veil, trama, and pileipellis, respectively.

## Figures and Tables

**Figure 1 jof-11-00123-f001:**
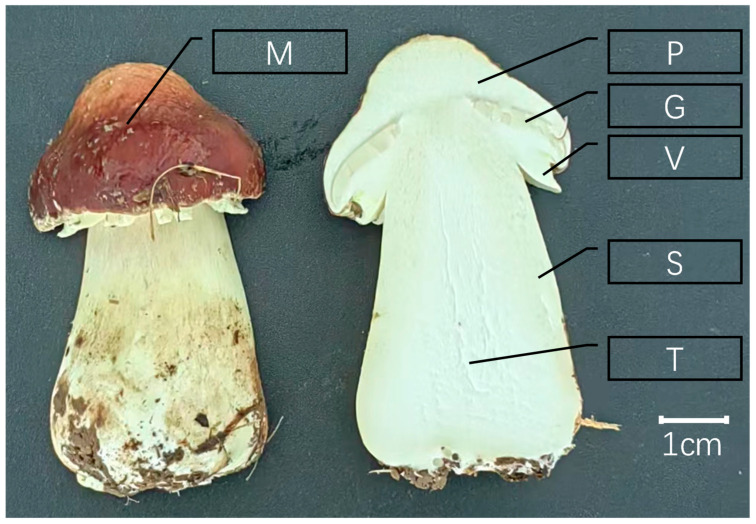
Different tissue samples of *S. rugosoannulata* for transcriptome sequencing, M: pileipellis; P: pileus; G: gill; V: veil; S: stipe; T: trama.

**Figure 2 jof-11-00123-f002:**
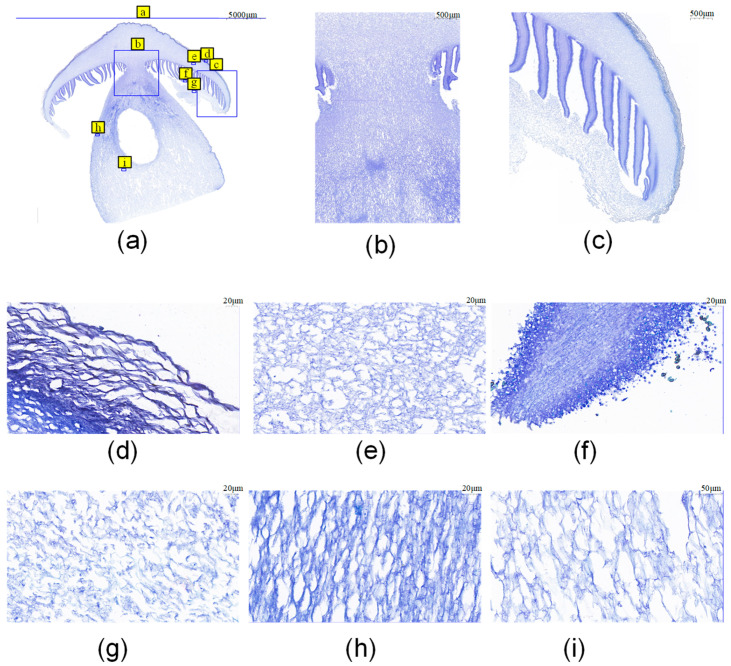
Observation of fruiting body structure of *S. rugosoannulata*; (**a**) The overall structure of *S. rugosoannulata* fruiting body, the letters and blue boxes indicate the position of the a–i structure in the fruiting body; (**b**) the structure of the connection between the stipe and the pileus; (**c**) pileipellis, pileus, and veil junction structure; (**d**) pileipellis structure; (**e**) pileus structure; (**f**) gill structure; (**g**) veil structure; (**h**) stipe structure; (**i**) trama structure.

**Figure 3 jof-11-00123-f003:**
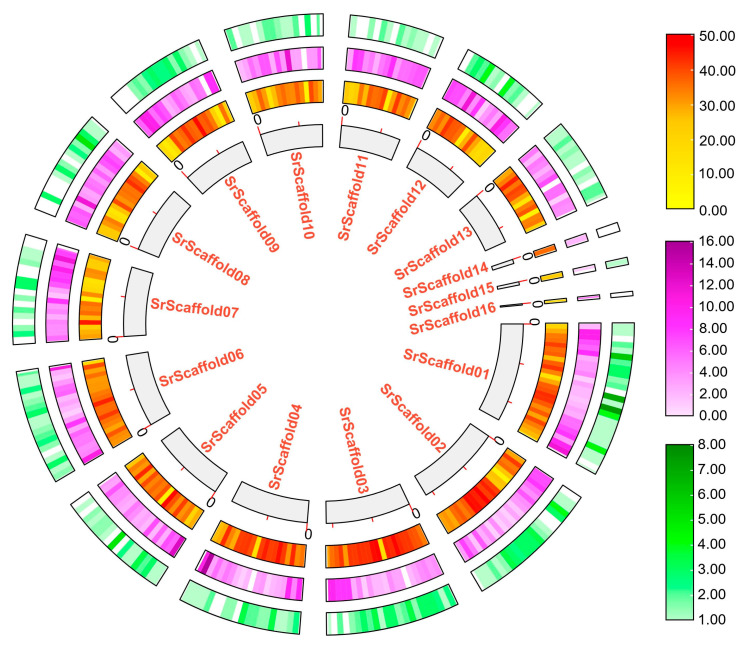
Characteristics of new gene annotation in the *S. rugosoannulata* genome; from the inner circle to the outer circle, the genome length, the gene density of the genome (v.1.0), the new gene density of the genome (v.2.0), the transcription factors, and the protein kinases found in the genome (v.2.0).

**Figure 4 jof-11-00123-f004:**
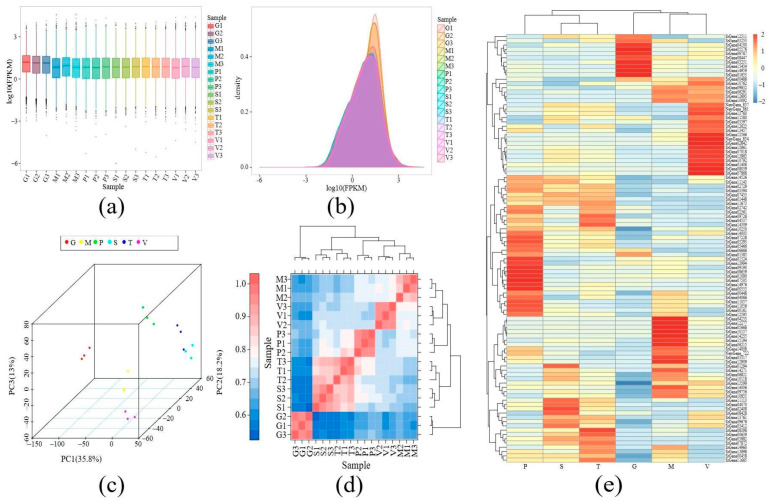
The relationship of the transcriptome in different tissues of *S. rugosoannulata*. (**a**) represents the FPKM density distribution comparison map of different tissues; (**b**) represents the FPKM box plot of different tissue samples; (**c**) represents the principal component analysis diagram, with different coordinates representing different principal components, the percentage representing the contribution value of the corresponding principal component to the sample difference, each point representing a sample, and different groups of samples represented by different colors; (**d**) represents the expression correlation heat map of different tissue samples; (**e**) represents the 100 genes with the highest expression in fruiting body tissues.

**Figure 5 jof-11-00123-f005:**
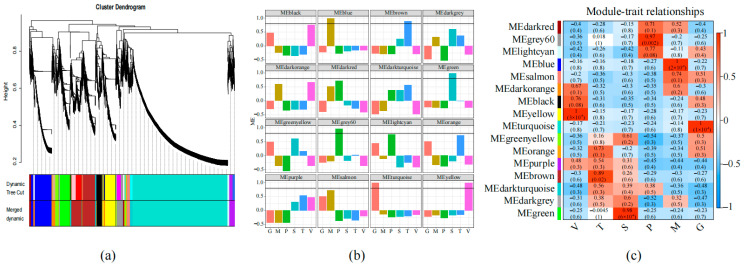
The relationship of the transcriptome in different tissues of *S. rugosoannulata*; (**a**) WGCNA module tree diagram, and different gene modules represented by different colors; (**b**) 16 co-expressed gene modules identified by WGCNA. The histogram was generated based on the ME values of all 16 modules, *X*-axis, different tissues, *Y*-axis, and ME values, and different groups of samples represented by different colors; (**c**) the correlation between modules and organizations. The correlation heat map of co-expressed gene modules with 6 tissues has a numerical Pearson correlation coefficient (**upper**) and corresponding *p*-value (**lower**). The color scheme, from red to white to blue, indicates the level of correlation from high to low.

**Figure 6 jof-11-00123-f006:**
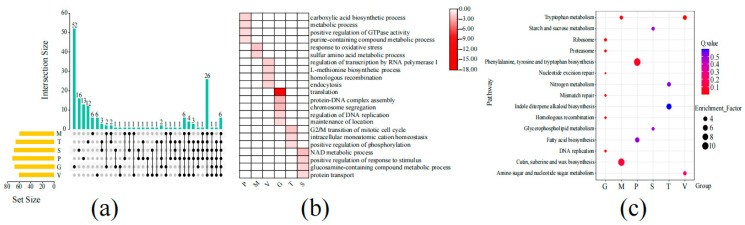
Biological process and KEGG analysis of genes in different tissues of *S. rugosoannulata*; (**a**) upset diagram of GO enrichment (biological process, BP) in different tissues; (**b**) main GO enrichment (BP) of each tissue gene base; (**c**) KEGG enrichment of each tissue gene base.

**Figure 7 jof-11-00123-f007:**
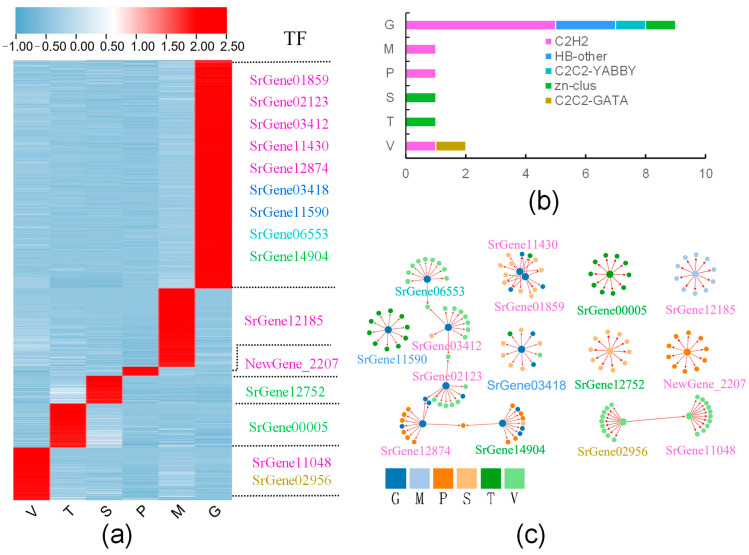
Tissue-specific expression of transcription factors in different tissues of *S. rugosoannulata*; (**a**) Specific expression of transcription factors in different tissues, wherein the gene name of the same color represents the same gene family and the family information is the same as in (**b**); (**b**) gene families and numbers of transcription factors specifically expressed in different tissues; (**c**) top 10 regulatory genes of specific expression of transcription factors in different tissues.

**Figure 8 jof-11-00123-f008:**
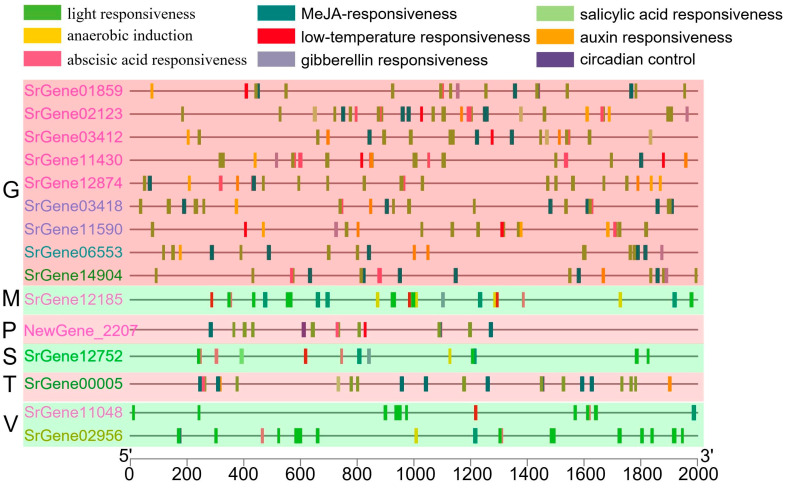
Analysis of cis-acting elements of transcription factors specifically expressed in different tissues of *S. rugosoannulata*.

**Table 1 jof-11-00123-t001:** Summary of the new *S. rugosoannulata* genome annotation.

Class	v.1.0	v.2.0
Number of transcripts	15,726	20,631
Number of genes	15,726	17,938
Mean mRNA length (bp)	1236.64	1286.43
Number per gene Genes with GO terms	6779	8369
Transcripts with KEGG terms Genes with functional	4063	6315
Number per gene Genes KOG system	4231	4675

## Data Availability

The original contributions presented in this study are included in the article. Further inquiries can be directed to the corresponding authors.

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
