# Peer review of "Analysis of Gene Regulatory Network and Transcription Factors in Different Tissues of the Stropharia rugosoannulata Fruiting Body"

_jof, 2025, doi:10.3390/jof11020123_

Round 1

Reviewer 1 Report

see an attached file

see an attached file

Author Response

We have responded to your review comments point by point, the content of the response Please see the attachment.

Reviewer 2 Report

This study investigates the regulatory networks of genes involved in the development of various tissues, as well as their constituent transcription factors, in the fruiting body of the mushroom Stropharia rugosoannulata. This research may contribute to the basis for elucidating the molecular mechanisms of fruiting body development in edible mushrooms. The dataset obtained in this study can serve as an important resource for the discovery and characterization of genes that confer beneficial traits to mushrooms. Furthermore, this research may provide guidance for improving S. rugosoannulata cultivation techniques.

The results and discussion are clearly described. The conclusion briefly states the main findings of the study.

A criticism of this manuscript is the lower resolution of Figures (3-8) and Figure S1; the results obtained are more difficult to monitor. Line 242: check if Figure 2 matches the given statement.

Author Response

(The authors gave the same response as above.)

Reviewer 3 Report

The manuscript ” Analysis of Gene Regulatory Network and Transcription Factors in Different Tissues of Stropharia rugosoannulata Fruiting Body” by Lu et al.  utilizes with skill many computational programs to analyze the RNA-seq libraries prepared from six different fruiting body tissues. The RNA data is also used to improve the published whole genome sequence QGU27 of S.rugosoannulata.  In spite of the skillful use of different computer programs to reveal the specific expression/coexpression of structural genes and transcription factors with their networks in different fruiting body tissues, one remains to miss comparisons with the data already published about genetic regulation of fruiting body differentiation in other basidiomycetes for instance in the following publications and several others.

Ohm, A et al. 2011. Transcription factor genes of Schizophyllum commune involved  in regulation of forming fungi (Agaricomycetes). doi: 10.1128/msystems.01208-23. Epub 2024 Feb 9

Kües U, Navarro-González M. 2015. How do Agaricomycetes shape their fruiting bodies? 1. morphological aspects of development. Fungal Biol Rev 29:63–97. doi:10.1016/j.fbr.2015.05.mushroom formation. Molecular Microbiology 81: 1433-1445.

Nagy et al. 2023. Lessons on fruiting body morphogenesis from genomes and transcriptomes of Agaricomycetes. Stud Mycol 104:1–85. doi: 10.3114/sim.2022.104.01

Földi et al.2024. Snowball: a novel gene family required for developmental patterning of fruiting bodies of mushroom-forming fungi (Agaricomycetes). doi: 10.1128/msystems.01208-23. Epub 2024 Feb 9

The following things need to be improved.

In general the sections (Fig. 2) of the fruiting body are not very informative. Lucky that there is Fig. 1 with the labels of different tissues studied. Lignification is the wrong word, there is no lignin is fungal cell walls, the authors probably mean thickening of cell wall- but no genes involved in cell wall synthesis were identified in different tissues. The sections of fruiting body also suggest “voids” openings in trama and veil, which could result from autophagy (autolysis of hyphae) but no genetic data pointing to this direction is given in the manuscript.

Results lines 214-215” The outermost layer of the gill is characterized by stromatocytes arranged in dense configurations, accompanied by the ejection of spores”/ The sentence is not right , stromatocytes separates gills and basidiospores are ejected from basidia between stromatocytes.

Chapter 3.2. “Genome re-annotation of S. rugosoannulata” is not explaining clearly how the new version v.2.0 was built up from v1.0 version QGU27.

Results lines 228-231 could read better: To investigate the key metabolic pathways in different fruiting body tissues of S. rugosoannulata S18, high-throughput RNA sequencing of pileus, pileipellis, gill, veil, stipe, and trama (Fig.1) was conducted, leading to a total of 18 RNA-seq libraries, three from each tissue.

In Results and Discussion several same subtitles are used as in Results, which indicates that the analyses presented in Discussion would suit better in Results. From the text of Results it is difficult to fish out the most significant achievements although the included Figs are informative.

Discussion should emphasize the new achievements obtained, compare with similar data obtained from previously investigated basidiomycetes, and state how the present results could be helpful in cultivation of Stropharia rugosoannulata.

See my statement for the authors

Author Response

(The authors gave the same response as above.)

Round 2

Reviewer 1 Report

The authors have taken all comments into account and improved the discussion of the results.

It's a real pity that other cis-element databases didn't work out to find something interesting. Plants and basidiomycetes are so far from each other in evolutionary terms, and it is likely that fungi have developed many additional cis-elements and transcription factors during their evolution. However, the authors have shown that plant databases are being used in some studies on fungi. The addition of information on hormones to the Discussion section was very interesting. An article can be published.

Minor comments:

Line 570 Fugal immunomodulatory protein in Flammulina velutipes and hat gene in Cordyceps militaris.

What is Fugal? fungal I think.

Line 579. Replace “monkey head mushroom” to “Pleurotus ostreatus” please.

Line 582. Ji et al., not  JI et al.

Line 585. Li et al., not LI et al.

Author Response

The authors have taken all comments into account and improved the discussion of the results.

It's a real pity that other cis-element databases didn't work out to find something interesting. Plants and basidiomycetes are so far from each other in evolutionary terms, and it is likely that fungi have developed many additional cis-elements and transcription factors during their evolution. However, the authors have shown that plant databases are being used in some studies on fungi. The addition of information on hormones to the Discussion section was very interesting. An article can be published.

Response: We thank you again for your comments and constructive suggestions, which are valuable for improving our future research direction.

Comments 1: Line 570 Fugal immunomodulatory protein in Flammulina velutipes and hat gene in Cordyceps militaris.

What is Fugal? fungal I think.

Response 1: Thank you again for your comments, due to our carelessness, caused you confusion, we refer to this paper, found here for Fugal immunomodulatory protein, and the manuscript has been revised.

Comments 2: Line 579. Replace “monkey head mushroom” to “Pleurotus ostreatus” please.

Response 2: Thank you again for your comments, due to our carelessness, caused you confusion, we refer to this paper, found here for Hericium erinaceus mycelia, and the manuscript has been revised.

Comments 3: Line 582. Ji et al., not  JI et al.   Line 585. Li et al., not LI et al.

Response 3: Thank you again for your comments, due to our carelessness, caused you confusion, We have changed JI et al. to Ji et al. on line 582 and LI et al. to Li et al. on line 585. respectively in the manuscript.

Reviewer 3 Report

The authors have done the requested corrections and additions

No addiotional comments

Author Response

Thank you so much for your review! We deeply appreciate your recognition of our research work.